# Self-DNA Early Exposure in Cultivated and Weedy *Setaria* Triggers ROS Degradation Signaling Pathways and Root Growth Inhibition

**DOI:** 10.3390/plants12061288

**Published:** 2023-03-13

**Authors:** Alessia Ronchi, Alessandro Foscari, Giusi Zaina, Emanuele De Paoli, Guido Incerti

**Affiliations:** 1Department of Life Sciences, University of Trieste, via Giorgieri 5, 34100 Trieste, Italy; 2Department of Agrifood, Environmental and Animal Sciences, University of Udine, via delle Scienze 206, 33100 Udine, Italy

**Keywords:** self-DNA inhibition, extracellular DNA, plant abiotic stress response, real-time qPCR, pest and weed control

## Abstract

The accumulation of fragmented extracellular DNA reduces conspecific seed germination and plantlet growth in a concentration-dependent manner. This self-DNA inhibition was repeatedly reported, but the underlying mechanisms are not fully clarified. We investigated the species-specificity of self-DNA inhibition in cultivated vs. weed congeneric species (respectively, *Setaria italica* and *S. pumila*) and carried out a targeted real-time qPCR analysis under the hypothesis that self-DNA elicits molecular pathways that are responsive to abiotic stressors. The results of a cross-factorial experiment on root elongation of seedlings exposed to self-DNA, congeneric DNA, and heterospecific DNA from *Brassica napus* and *Salmon salar* confirmed a significantly higher inhibition by self-DNA as compared to non-self-treatments, with the latter showing a magnitude of the effect consistent with the phylogenetic distance between the DNA source and the target species. Targeted gene expression analysis highlighted an early activation of genes involved in ROS degradation and management (*FSD2*, *ALDH22A1*, *CSD3*, *MPK17*), as well as deactivation of scaffolding molecules acting as negative regulators of stress signaling pathways (*WD40-155*). While being the first exploration of early response to self-DNA inhibition at molecular level on C4 model plants, our study highlights the need for further investigation of the relationships between DNA exposure and stress signaling pathways by discussing potential applications for species-specific weed control in agriculture.

## 1. Introduction

Extracellular self-DNA inhibitory effect is a recently discovered phenomenon caused by the accumulation of fragmented DNA in decomposing plant litter, which significantly reduces conspecific seed germination and plantlet growth while barely affecting heterospecifics. By exerting such species-specific inhibitory effects in a concentration-dependent manner, self-DNA could represent a further explanatory process underlying negative plant–soil feedback [1,2,3]. This autotoxic effect has also been tested and verified in other organisms from different kingdoms and environments, including bacteria, fungi, algae, protozoa, and insects, suggesting a more general biological process [4,5]. The discovery bears important implications for plant ecology, as the accumulation, persistence, or removal of DNA in the soil—depending on the environment, soil characteristics, and weather conditions—could play a fundamental role in determining biodiversity levels and patterns in different ecosystems [6]. Self-DNA in soil might also function as a signaling molecule for self-damage recognition, triggering plant resistance against environmental stresses and dangers such as pathogen infection, herbivore feeders, and intraspecific competition [7,8]. Moreover, self-DNA inhibitory effects can have significant applications for the development of novel pharmaceuticals [9], as well as in agriculture for pest and weed control [10].

In this context, we present the results of an experiment carried out on two target species belonging to the genus *Setaria*: *S*. *italica* (L.) P. Beauvois and *S. pumila* (Poir.) Roem. and Schult. We purposely chose these two species for three main reasons. First, the genus *Setaria* can be considered a model plant genus for C4 metabolism [11], with an increasing number of published studies addressing its genetics and genomics [12]. Second, the two species provide, for the first time, an interesting case study to test the species-specificity of self-DNA inhibition in cultivated vs. weed congeneric species of global relevance. Indeed, *S. italica* (the foxtail millet) is a crop used worldwide for human (in East Asia and Middle East) and livestock (Europe and North America) feeding, while *S. pumila* (the yellow foxtail) is a weed of global concern with a severe impact on dairy pastures, such as in New Zealand [13] and in Switzerland [14], and on cereal crops, as in the United States and Canada [15]. Therefore, such a test could provide interesting insight on the application perspectives of the self-DNA inhibition principle as species-specific weed control [16]. Third, the availability of the sequenced genome of *S. italica* [17] also allows for the assessment of the species response to conspecific and congeneric DNA exposure at a genetic level. Here, in order to assess whether the self-inhibition principle still holds for the two target species, we exposed seedlings to treatment with DNA extracts from four different sources (i.e., conspecific, congeneric, plant heterospecific from *Brassica napus* L., and animal from *Salmon salar* L.) at three different concentrations in a cross-factorial experiment. 

The phenomenological evidence on self-DNA inhibition has been repeatedly reported, with an inhibitory effect in plants consistently observed after an exposure time ranging between 3 days and 4 weeks [1,18], correlated to the taxonomic distance between the receiver and the source DNA species [1,4,18]. Differently, the underlying mechanisms at cellular and molecular levels are not yet fully clarified, although they have been explored by some previous papers [18,19,20,21,22], mostly referring to the time window preceding the inhibition observation. With the exception of the biochemical and epigenetic tests carried out after 5 days of exposure in *Lactuca sativa* plants by Vega-Munoz et al. [19], the studies addressing the inhibition mechanisms refer to an observation time spanning between 30 min and 16 h after exposure [18,19,20,21,22,23]. In particular, Duran-Flores and Heil [18] observed the activation of Mitogen-Activated-Protein-Kinases (MAPK) at 30 min, H_2_O_2_ production at 2 h, and extra-floral nectar production at 24 h post-treatment in Lima bean (*Phaseolus lunatus*) and maize (*Zea mays*), suggesting that self-DNA acts as a damage-associated molecular pattern (DAMP), inducing early immunity-related signaling responses. Accordingly, a reduced rootlet growth would result in a response to the energetic cost of the immunity response [20]. Further previous evidence on the same species includes an increase in cytosolic flux of Ca^2+^ after 30 min, associated with a concentration-dependent plasma transmembrane potential (Vm) depolarization at 2 h [21], later confirmed by the same authors on tomato (*Solanum lycopersicum*) leaves, coupled to the opening of K^+^ channels at 50 min, and followed by ROS production after 180 min [8]. Moreover, 1 h exposure to self-DNA elicited an alteration of the transcriptomic profile involving several genes related to Ca^2+^ signaling, ROS scavenging, and ion homeostasis [8]. A very recent metabolomic profiling during self-DNA exposure, between 1 to 15 h in *A. thaliana* plantlets [22], highlighted a striking, progressive accumulation of nucleobases, ribonucleosides, dinucleotide, and trinucleotide oligomers—in particular, cyclic AMP and GMP, as well as N6 methylated adenosine. Such a finding was interpreted as an indication of RNA degradation and a lack of disposal or recycling with consequent metabolic impairment based on previous findings of a dramatic reduction in gene expression along the same time frame, which was observed on the same model plant by Chiusano et al., 2021 [23]. However, this latter study, a whole-plant transcriptomic profiling, highlighted a remarkable pattern of differential gene expression across treatments (self-DNA vs. non-self-DNA) and timings (1, 8, and 16 h), with a significant differential expression of several pools of genes, noteworthy among which were those responsive to abiotic stress under self-DNA exposure. This was mostly evident after 1 h exposure, and then, it was apparently released after 8 h.

Therefore, in this study, we test if the evidence reported by Chiusano et al. [23] still holds for the two *Setaria* species over the time window, spanning between 1 and 3 h, since exposure. We present the results of a real-time qPCR analysis of seven genes known to respond to drought, osmotic, oxidative, and thermic stress in *S. italica*. While the effects of self-DNA on congeneric species were previously investigated [18], this is the first study comparatively and simultaneously testing self-DNA inhibition on a cultivated and an invasive congeneric species. Our hypothesis is that the species-specificity of self-DNA inhibition still holds when tested on phylogenetically related species, even on weed plants that are expected to be more resistant to allelopathic effects. From an application perspective, evidence of species-specific self-DNA inhibition on the invasive weed *S. pumila* but not on the cultivated species *S. italica* could provide promising data for innovative weedicide treatments in agriculture. Finally, from a pure science perspective, our study contributes to the ongoing investigation on the molecular mechanisms underlying the observed phenomenon of self-DNA inhibition, with a particular focus on the early response to exposure, at gene expression scale. In this respect, we hypothesize that early exposure to self-DNA elicits molecular pathways known to be responsive to abiotic stressors.

## 2. Results

### 2.1. Root Elongation in Response to DNA Exposure

Our cross-factorial experiment (Figure 1) showed a significant effect of target species, DNA source, concentration, and their interactions on the root elongation of *S. italica* and *S. pumila* seedlings (Table 1). Both target species showed significantly lower root elongation when exposed to self-DNA, as compared to all other treatments, and consistently across all the tested concentration levels (Figure 1, Appendix A). Moreover, DNA from congeneric species produced higher inhibition as compared to DNA from other heterospecifics, especially when comparing congeneric vs. *S. salar* DNA effects, although this was more evident at the highest DNA concentration (Figure 1, Appendix A). Such a pattern was consistent with the significant effects of the D and D × C terms in the ANOVA model (Table 1, Appendix A). Sensitivity to treatments was species-specific, as indicated by the significant S × D term in the ANOVA model (Table 1, Appendix A), with *S. italica* showing root growth inhibition at all tested self-DNA concentration levels, while *S. pumila* rootlet was not inhibited at the lowest self-DNA concentration (Appendix A).

### 2.2. Expression of Abiotic Stress Responsive Genes in Response to Self-DNA 

Mean extracted RNA yields were 1207 ng per root mg (*S. italica*) and 1125 ng per root mg (*S. pumila*). RNA integrity was satisfactory, with RIN values ranging between 5.00 and 6.60. Mean cDNA yields (DNA-50) from 1 µg of RNA were 32 µg (*S. italica*) and 30 µg (*S. pumila*). The pool of genes selected for the real-time qPCR experiment showed a very similar expression pattern for both target species (Figure 2), although *S. pumila* generally presented the highest response level (the range of fold change in gene expression was 0.195–2.305 in *S. italica* and 0.234–2.960 in *S. pumila*, Appendix A). 

In particular, the target genes *FSD2*, *ALDH22A1*, *ALDH7B1*, and *CSD3*, respectively, were responsive to drought, osmotic, oxidative and cold stress (*FSD2*), osmotic and oxidative stress (*ALDH22A1* and *ALDH7B1*) as well as osmotic, oxidative, and cold stress (*CSD3*), which were upregulated in both species at both observation times (Figure 2) and substantially consistent with their known response to abiotic stressors. In the cases of *FSD2* and *ALDH22A1*, mean ΔCq values were also significantly different from the respective controls, while in the case of *CSD3*, the expression values were significantly different from the control only in *S. pumila*, with an increase, with time, of its expression levels from 1 h to 3 h (Appendix A). 

The genes *WD40-144* and *WD40155*, respectively responsive to osmotic, oxidative, and cold stresses (*WD40-144*) and to drought, osmotic, oxidative, and cold stress (*WD40155*), showed a peculiar expression pattern, were characterized by a generalized downregulation in response to self-DNA not previously reported for other abiotic stressors (Figure 2. Specifically, *WD40-155* mean ΔCq values resulted in significantly different values from the control at each exposure time and for both species, while also showing a significant decrease in its expression levels with time (Appendix A). Finally, *MPK17*, normally involved in dehydration and hyper-osmotic stress, was initially upregulated in both species (mean ΔCq at 1 h was significantly different from the control, Appendix A), as previously reported for other abiotic stressors, and then, it later showed a significant decrease in its expression levels in both species (Figure 2).

## 3. Discussion

### 3.1. Species-Specificity of Self-DNA Inhibitory Effect

A self-inhibition by fragmented extracellular DNA, mostly for fragment size between 50 and 1000 bp, has been reported in previous studies as dependent on the concentration of DNA in the growing substrate and on the phylogenetic distance between the DNA source and the receiver species [1,4,18]. Since its discovery, the magnitude of self-DNA inhibition was related to the species-specificity of the molecular agent. In particular, in Mazzoleni et al. [1], a stronger effect of conspecific DNA is highlighted, as compared to heterospecific DNA, with intermediate magnitude of the inhibition when the target and the DNA source species belong to the same taxonomic family. Duran-Flores and Heil [18] confirmed the species-specificity of self-DNA, showing that common bean (*Phaseolus vulgaris*) root growth was strongly inhibited by self-DNA, weakly inhibited by congeneric DNA (*Phaseolus lunatus*), but substantially unaffected by heterologous DNA from acacia (*Acacia farnesiana*), indicating that the species-specificity of the self-DNA effect still holds at the infrageneric level. Along this line, we tested the species-specificity of self-DNA inhibition in congeneric species with the novelty of investigating a cultivated (*Setaria italica*) and a weedy, invasive species (*Setaria pumila*), with the latter expected to be more resistant to environmental stressors [24,25,26,27,28]. In our cross-factorial experiment, the absence of detectable effects of *S. salar* DNA and a marginal effect of heterospecific DNA from *B. napus* on *Setaria* rootlets are fully consistent with the above-mentioned previous findings, confirming the absence of inhibition in the case of species exposed to DNA from phylogenetically distant species, while still showing a weak, marginal concentration-dependent inhibition exerted by heterologous plant DNA at a supra-familiar phylogenetic distance [1,4,18]. Taken together, our results provided confirmatory evidence on the absence of a substantial effect of extracellular DNA, from phylogenetically distant species, on the root elongation of target plants. 

Considering, with more detail, our results on the two congeneric target plants and the effects cross-factorially exerted by exposure to their DNA, the observed pattern of significant inhibition of root elongation was fully consistent with previous findings and our expectations. In particular, the inhibitory effect of conspecific DNA, on both *S. italica* and *S. pumila* root growth, was significantly higher than the one exerted by congeneric DNA at the same concentration levels, highlighting the species-specificity of the self-DNA effect at infra-generic level. The magnitude of self-DNA inhibition observed in our experiment is also consistent with that previously observed at similar concentration levels for different plant species [1,4,18]. At the lowest concentration level (2 ng/µL), only the *S. italica* seedlings were significantly inhibited by self-DNA, thus providing support to the general hypothesis of a higher susceptibility of cultivated species, compared to invasive weeds, towards environmental stress factors [24,25,26,27,28]. Accordingly, *S. pumila* DNA at 10 ng/µL, besides inhibiting conspecific seedlings, also showed a marginal inhibitory effect on congeneric (*S. italica*) seedlings, although in this latter case, the treatment vs. control comparison produced a borderline *p*-value. Therefore, in the context of species-specific biological control, our study highlights the promising role of *S. pumila* DNA as a potential species-specific weedicide in analogy to its previously suggested use as a species-specific pesticide [10,29,30,31]. However, upscaling tests in an open field are obviously required in order to clarify the persistence of extracellular DNA and the reliability of its self-inhibitory effects under more realistic conditions, as well as the possible interference with cultivations of phylogenetically related crops.

### 3.2. Expression of Abiotic Stress Responsive Genes after Self-DNA Exposure

#### 3.2.1. Drought and Dehydration Stress 

Drought stress in plants means that transpiration or evaporation exceeds water uptake in plants [32], and it is closely intertwined with dehydration, as the first event during drought stress is the loss of water from the cell [33] with consequent reduction in water potential and turgor [34]. Drought is considered one of the most important environmental stresses in agriculture [35]. It leads to physiological and morphological adaptations to reduce evapotranspiration, such as decreased leaf area or leaf folding, ABA-mediated stomatal closure, increased leaf thickness, and enlargement of the root system, together with plant growth and productivity decrease [36,37].

From a molecular point of view, several genes are activated and involved in response and signaling pathways in *S. italica* under drought conditions, among which we selected *FSD2*, *WD40-155*, and *MPK17-1* [38,39,40]. *FSD2* encodes an iron–superoxide dismutase (FeSOD), and its expression level is reported to decrease (relative to control) after 1 h of drought stress and to significantly increase (fold change ≅ 5) and peak after 4 h [38]. In our real-time qPCR analysis, *FSD2* was also significantly upregulated (fold change ranging between 2 and 3) at both exposure times (1 h and 3 h) in both species. Comparatively, this result suggests an earlier activation in response to self-DNA as compared to drought stress, although a direct quantitative comparison is not straightforward as it is possibly biased by the different stress nature and intensity between our experimental conditions and those of the reference study. However, since SODs are known to play a crucial role, by the dismutation of O_2_^−^ radicals, in the protection against oxidative damage [41], our result is consistent with an enhanced early superoxide production under self-DNA exposure. This finding is also consistent with the enhanced expression of genes related to anti-oxidant activity found in *A. thaliana*, after 1 h exposure to self-DNA, in the transcriptomic study by Chiusano et al. [23], among which 5 peroxidases and, remarkably, the Fe superoxide dismutase 1 (*FSD1*) functionally analogue to our target gene. Interestingly, a very recent work [42] showed higher levels of O_2_ˉ and H_2_O_2_ in rice (*Oryza sativa* L.) roots, after 7 days of exposure to self-DNA, compared to the unexposed control, although the experimental timing prevents us from assessing if this corresponded to a prolonged ROS production or a decreasing trend after an earlier peak. However, the authors also observed a down-regulation of ROS-scavengers encoding genes at the same time-point, which was interpreted, there, as a signal of decreasing, but they were still high in ROS content and informative of a preceding cytotoxic redox state. Differently, Vega-Muñoz et al. [19], in a qPCR assay after whole plant total RNA extraction, reported that antioxidant genes (superoxide dismutase/SOD, catalase/CAT, and phenylalanine ammonia lyase/PAL) were up-regulated in a concentration-dependent manner after 5 days of self-DNA exposure in lettuce (*Lactuca sativa* L.). The function of ROS production and scavenging, along the response dynamics to self-DNA, cannot be clarified by summing up our and previous findings, due to several experimental differences, including the target species and plant organ, experimental timing, and exposure dose. However, both cited studies suggest a long-term role in self-DNA stress management. At an earlier term, ranging between 1 and 3 h, our observation of ROS activation is consistent with the studies of Barbero et al. [8] and Duran-Flores and Heil [18]. In both cases, peroxidase activity was found, respectively, by fluorescent dye and enzymatic assay in the chloroplasts of tomato leaves and in lima bean leaves 3 and 2 h after exposure to self-DNA. However, both were, previously, critically used to mechanically damage the leaf material before or after the exposure to self-DNA, potentially exacerbating the production of H_2_O_2_, which is a well-known end product of the DAMP cascade [43].

*WD40-155* encodes the WD repeat-containing protein DWA2 and was found to be upregulated during dehydration stress at 1 and 3 h, reaching its peak expression at 3 h and, then, decreasing at a longer term [39]. We observed the opposite response pattern for this gene, with a significant downregulation in all tested conditions. Its trend in *S. pumila* even suggests an increasing downregulation with time. Since DWA2 protein is known as a negative regulator of ABA signaling in *A. thaliana* [44], it could be inferred that such a signaling pathway plays an important role during early response to self-DNA exposure, as already pointed out by Chiusano et al. [23], showing an early upregulation of genes related to ABA and jasmonic acid at 1 h of self-DNA treatment. In fact, ABA is a very important stress hormone in plants, accumulated in response to stress conditions in different organs and able to initiate a cascade of signal transduction pathways that regulate stomatal aperture and expression of genes involved in resistance to environmental stresses [45]. It also interacts with the jasmonic acid (JA) and salicylic acid (SA) signaling pathways, and it is reported to be involved in signaling crosstalks between biotic and abiotic stress responses [46]. However, its most important function is the regulation of plant water balance and osmotic stress tolerance [45]. Accordingly, in *Setaria*, the negative regulator of ABA signaling, DWA2 protein, is downregulated for prolonged drought conditions, while self-DNA exposure seems to trigger an earlier onset of ABA signaling cascade. 

Finally, *MPK17* encodes a mitogen-activated protein kinase that showed the highest expression level (around 6-fold induction) after 1 h of dehydration stress in a tolerant cultivar of *S. italica* and an earlier, but lower, peak in a non-tolerant cultivar; then, it released at 3 h in both cultivars [40]. Consistently, in our analysis, *MPK17* is firstly upregulated after 1 h of self-DNA exposure and, then, significantly downregulated after 3 h in both species. Interestingly, the upregulation at 1 h is perfectly consistent with the MAPKs activation previously described in common bean after 30 min of exposure to self-DNA [18], which, in turn, can be triggered by ROS production [47]. Moreover, a recent genetic study [48] in rice (*Oryza sativa* L.), a species phylogenetically closely related to *S. italica* [49], highlighted that the downregulation of *MPK17* enhances Xa21-mediated resistance to the bacterial *Xanthomonas oryzae* pv. *Oryzae* (Xoo). The downregulation of *MPK17* at 3 h, in our analysis, could be related to the plant immunity response to self-DNA, which is already hypothesized to function as a damage-associated molecular pattern (DAMP), indicating self-damage and triggering self-specific immunity induction [8,18,19]. Finally, the downregulation of *MPK17* seems to affect plant morphology, significantly reducing growth, development, and reproduction [48].

#### 3.2.2. Osmotic and Oxidative Stress

High salt concentration in soil alters plant performance by causing metabolic damage, ion toxicity, secondary oxidative stress, and osmotic stress, and it induces gene expression alterations fitting an efficient salt stress response [50]. Oxidative stress, which can be triggered by different severe environmental stress factors, is associated with an excessive production and accumulation of ROS, toxic molecules that can cause damage by lipid peroxidation, affecting nucleic acids and protein oxidation, which promote programmed cell death [51]. Among the genes involved in salinity and osmotic stress response in *S. italica*, we selected *ALDH22A1*, *ALDH7B1*, *CSD3*, and *WD40-144* [38,39,52] in addition to the three genes described above and already selected as responsive to drought and dehydration (*FSD2*, *WD40-155*, *MPK17*) [38,39,40]. *ALDH22A1* and *ALDH7B1* encode aldehyde dehydrogenases (ALDHs), enzymes known to reduce oxidative stress, catalyzing the oxidation of a wide range of aldehydes into corresponding carboxylic acids, detoxifying cellular ROS, and/or reducing lipid peroxidation [52,53]. During salinity stress in *S. italica*, *ALDH22A1* is upregulated after 1 h, reaching its peak after 6 h, while *ALDH7B1* results upregulated only after 6 h, suggesting a later activation [52]. Our analysis in response to self-DNA substantially highlighted the same pattern, with *ALDH7B1* expression not significantly different from the control at 1 and 3 h, as well as a significant upregulation at 1 and 3 h for *ALDH22A1*. *CSD3* encodes a Cu–Zn superoxide dismutase, and its expression level is reported to decrease after 1 h and then increase after 4 h [38]. This is in line with the non-significant changes in expression level of *CSD3* after 1 and 3 h exposure to self-DNA in *S. italica*. Interestingly, *CSD3* was significantly upregulated in *S. pumila*, already, at 1 and 3 h, indicating an expression progressively increasing with time, as reported for other invasive plants, which show superior tolerance to drought and salinity stress in connection to a more efficient upregulation of SODs [54]. Then, this might be related to a higher stress resistance of the weedy invasive *Setaria* species as compared to the cultivated one [24,25,26,27,28]. In addition, *S. pumila* showed a wider range of gene expression variation relative to the control, as compared to *S. italica* (Appendix A), possibly indicating a more rapid and intense response to stress onset. 

*WD40-144* was found to be strongly upregulated under salt stress after 1, 3, and 6 h [39]. On the contrary, in our analysis the expression level of this gene did not vary significantly among treatments, indicating that it is not likely involved in the response to self-DNA, at least at an early stage. About the genes already mentioned in the previous subsection, *WD40-155* presents an oscillating pattern in response to salt and osmotic stress, being slightly upregulated at 1 h, but not at 3 h, and then reaching its peak at 6 h [39], while it was significantly downregulated in response to self-DNA at 1 and 3 h. *MPK17*, which we mentioned in the previous subsection as being responsive to water stress in *S. italica*, is also reported as responsive to salt stress in other plant species. In *Arabidopsis thaliana* [55,56], it is transiently induced after 3 h of hyperosmolarity influencing the proliferation and cellular distribution of peroxisomes; in maize (*Zea mays* L.), both PEG and H_2_O_2_ treatment caused a decline in the expression of *ZmMPK17* in roots, correlated to increased Ca^2+^, and the lower peaks appeared at 24 and 3 h, respectively [57]. In our analysis, *MPK17* results significantly downregulated at 3 h; this is intriguingly consistent with the findings by Barbero et al. [21], reporting an increase in cytosolic Ca^2+^ concentration after early exposure to self-DNA in *Z. mays*. 

#### 3.2.3. Thermic Stress 

We considered the gene expression response to cold stress, a dangerous environmental stressor that can cause cell membrane damage and cell-cycle disruption, affecting plant germination, growth, development, and reproduction [58]. Among the genes involved in the early response to cold stress in *S. italica*, we had selected two SODs, *FSD2* and *CSD3* [38], already discussed above as also responsive to other abiotic stressors. Both genes were found upregulated in response to cold stress, which is consistent with the generalized enhancement of the SOD gene family in foxtail millet under stress conditions [38]. Treatments with self-DNA elicited, substantially, the same pattern, especially for the more tolerant weed *S. pumila*, with the only exception of *CSD3* in *S. italica* showing a non-significant upregulation. Given the prominent function of these two genes in the anti-oxidant response to several stress factors, including self-DNA exposure, the observed pattern does not provide useful insight about the relationships between the response to self-DNA and that to thermal stress. 

## 4. Materials and Methods

### 4.1. Root Elongation Experiments 

#### 4.1.1. Leaf Biomass Production for DNA Extraction

*S. pumila* seeds were collected in the field in Cadenazzo (Switzerland) in the late summer of 2020; seeds of *S. italica* (Indo American Hybrid Seeds (I) pvt. Ltd. Bangalore, India) and *B. napus* (not tanned Gordon variety, KWS Italy S.p.a.) were purchased from the Friulian Agricultural Club of Udine (Udine, Italy). Seeds of each species were imbibed with Milli-RO water for 24 h into in 50 mL lab grade tubes, and then, they were transferred to plastic saucers filled with a standard peat:perlite growing substrate, and they were kept there until germination. After germination, seedlings were transplanted in 8 cm pots (2 seedlings per pot) previously filled with the substrate. Plants of all three species were grown under controlled conditions (day T = 22 °C; night T = 20 °C; photoperiod = 12 h; relative humidity = 50%; PPFD = 600 µmol photons m^−2^ s^−1^) for 60 days. Then, the foliar biomass was harvested. 

#### 4.1.2. DNA Extraction and RNase Treatment 

Nucleic acid extraction from leaf material of *S. italica*, *S. pumila*, and *B. napus* was carried out by a modified Doyle and Doyle [59] protocol. For each extraction, 5 g of fresh leaves were grounded in liquid nitrogen and placed in a 50 mL Falcon tube containing the lysis solution composed of 20 mL CTAB (2.5%), 2 μL Proteinase K (20 μg/μL), and 200 μL β-mercaptoethanol (0.1%). The tube was incubated at 65 °C for 30 min and then transferred on ice for 10 min. To separate nucleic acids from cellular components (proteins, lipids, polysaccharides) and other interfering substances (polyphenols), 20 mL of the chloroform–isoamyl alcohol mixture (24:1) were added. The tube was stirred by inversion for 10 min and centrifuged for 30 min at 6800 rpm. Then, the aqueous supernatant fraction was gently pipetted out. Sodium acetate (3M, 1/10 starting volume) and pure 2-propanol (2/3 of the final volume) were added, followed by incubation at −20 °C for 1 h and centrifugation for 30 min at 6800 rpm. Liquid was discarded, and the residual pellet was washed with 2 mL of 80% ethanol twice. All traces of ethanol were removed by heat volatilization (37 °C for 10–15 min). At the end, the nucleic acid pellet was resuspended in 2 mL of sterile deionized water. To remove RNA, 20 µL of RNase A enzyme (10 mg/mL) was added to the tube and incubated for 1 h at 37 °C. A further precipitation step was performed by adding ammonium acetate (10 M, pH = 7, 1/3 starting volume) and 100% ethanol (2 final volume). The DNA pellet was washed with ethanol as described above. Finally, the DNA pellet was resuspended in 2 mL of sterile deionized water.

#### 4.1.3. DNA Treatment Solution Preparation

In order to replicate the molecular size observed in natural conditions and produced by chemical–physical degradation after plant debris decomposition [1], extracted DNA solutions (about 20 mL for each of the three plant species) were sonicated using the sonicator model UP200S (Hielscher, Teltow, Germany) for 4 min at full power, with alternating high and low-pressure cycles of 1 s. Commercial *Salmon salar* DNA solution (deoxyribonucleic acid from salmon sperm (Merck, Darmstadt, Germany) was already bought at low molecular weight, so it was not exposed to the fragmentation process. The fragment length distribution in all DNA solutions was assessed by 0.8% agarose gel electrophoresis. All DNA solutions were diluted at 2, 10, and 50 ng/µL to be used for treatments in the cross-factorial experiment. Limited to *Setaria* DNA, the treatment solutions were also ultra-purified with the AMPure XP system (Beckman Coulter, Brea, CA, USA), a paramagnetic bead SPRI (Solid-Phase Reversible Immobilization) technology generally used for the preparation of highly-pure genetic material [60,61], following the manufacturer’s recommendations.

All DNA solutions were quantified by fluorimeter Qubit 3.0 (Life Technology, Carlsbad, CA, USA), and the quality was assessed by spectrophotometer Nanodrop ND 1000 (Thermo Fisher Scientific, Waltham, MA, USA).

#### 4.1.4. Experimental Setup 

Seeds of *S. pumila* and *S. italica* were sterilized with a 20% sodium hypochlorite solution, thoroughly washed with sterile deionized water, and placed in Petri dishes (Vetrotecnica, Padova, Italy) over three sheets of filter paper (Grade 1 qualitative filter paper, Whatman, Maidstone, UK) soaked with 4 mL of sterile deionized water. Dishes were placed in a growth chamber under standard controlled conditions (22 ± 2 °C, 50% RH, 16 day and 8 night photoperiod) for 4/5 days. After germination, seedlings with radicle length between 2 and 5 mm were selected for each species and transferred in new Petri dishes (12 seedlings per dish) over filter paper soaked with 4 mL of either sterile deionized water (controls) or one of the DNA solutions (treatments) described in Section 4.1.3 and exposed for 4 days under the same previous standard controlled conditions. For the cross-factorial root elongation experiment, 3 replicated dishes were set up, plus 3 control dishes, for each target *Setaria* species and for each treatment combination of DNA source and concentration for a total of 78 experimental units (3 replicates × 2 species × 4 DNA sources × 3 concentration levels + 6 controls). 

At the end of the exposure phase, all the seedlings from each Petri dish were moved onto graph paper and photographed (Figure 3). The images obtained before and after the exposure were analyzed with the software ImageJ version 1.51 (https://imagej.nih.gov/ij, accessed on 7 March 2023, National Institutes of Health, Bethesda, MD, USA), and root elongation was calculated for each seed. Root elongation data, within each target species and treatment, were expressed as averages of the replicates (each calculated over the seeds in the dish) and as percentage of the corresponding control. 

### 4.2. qPCR Experiment for Abiotic Stress Responsive Genes

#### 4.2.1. Gene Selection and Primer Design

Since only the *S. italica* genome has been fully sequenced (Joint Genome Institute, USA, and Bijing Genome Initiative China), in this study, we used *S. italica* as a reference genome for *S. pumila* as well. There were 7 genes (*FSD2*, *ALDH22A1*, *ALDH7B1*, *CSD3*, *WD40-155*, *WD40-144*, *MPK17*) involved in *Setaria italica* signaling pathways that were responsive to abiotic stress and known to be up or downregulated within the first 6 h of exposure to the abiotic stressor in *S. italica* root that were selected from previous studies [38,39,40,52] as the reference gene coding for *RNA Polymerase II* [62]. As different genes are known to respond to several stress factors, we separately discuss all abiotic factors considered in this study (i.e., drought, osmotic, oxidative, and thermic stress), as previously suggested [50,63], to better investigate the potential connection between the expression response of the target genes after self-DNA exposure and their expression levels under a specific abiotic stress. 

Real-time qPCR primers were selected as follows: for the reference gene and the target genes, *ALDH22A1* and *ALDH7B1* we used the same primers proposed by the authors [53,63], as they met the analysis requirements for amplicon length, melting temperature, and position on the genomic sequence. For all the other target genes, instead, we proceeded to design the primers using the Primer3web v.4.1.0 software (ELIXIR Estonia), setting the following parameters: primer length (Min. 18; Opt. 20; Max. 24 bases), primer melting temperature (Min. 64 °C; Opt. 65 °C; Max. 66 °C), and amplicon length (130–210 bases). Inputs for Primer3 software were *S. italica* CDS (coding DNA sequence) of the target genes, available on the Phytozome database (Phytozome v.13, Joint Genome Institute, JGI, Berkeley, CA, USA). Primers were designed to be placed on two contiguous exons to detect genomic residual traces during controls with qualitative PCR or partly on one exon and partly on the following one to be able to amplify only retrotranscripted RNA sequences. Primers used in the present study are listed in Appendix A and were sourced from Sigma Aldrich (Rome, Italy). Eventually, we verified that the region amplified by the selected primers did not have high similarity with other sequences of the *S. italica* genome (through BLAST tool on Phytozome website) to prevent primers from amplifying unspecific targets. Moreover, we verified that there were no high similarities in the sequences of gene members belonging to the same family: in particular, *ALDH22A1* and *ALDH7B1*, as well as *WD40-155* and *WD40-144*. For this analysis, we utilized the Clustal Omega software (EMBL-EBI, Wellcome Genome Campus, Hinxton, Cambridgeshire, UK), which allows to find the best alignment among a given number of nucleotide sequences. Specific primer amplification was also verified on a retrotranscribed RNA for both *S. italica* and *S. pumila* through qualitative PCR (50 ng per cDNA sample, T annealing = 58 °C, 35 cycles using OneTaq Hot Start DNA Polymerase from New England Biolabs, Ipswich, MA, USA) and 2% electrophoresis agarose gel.

#### 4.2.2. Self-DNA Exposure 

Seeds of the two target species, *S. italica* and *S. pumila*, were prepared as described in Section 4.1.4. After germination, seedlings with radicle length between 5 and 10 mm were selected for exposure. Each seedling was placed on a Petri dish and exposed to 10 µL of 90 ng/µL ultra-purified self-DNA solutions (Section 4.1.3) by micro pipetting on the root apex. Petri dishes were closed with lids during exposure at room temperature to minimize evaporation of the treatment solution. There were 3 biological replicates (i.e., Petri dishes with 20 germinated seeds each) set up for each combination of target species and exposure time (1 and 3 h) plus 3 control replicates (dishes containing seedlings micro pipetted with deionized sterile water), for each species and time, for a total of 24 Petri dishes (3 replicates × 2 species × 2 exposure times + 12 controls). After undergoing the self-DNA treatment, seedling radicles were collected from each Petri dish, fresh-weighed, and stored at −80 °C. 

#### 4.2.3. RNA Extraction, Purification and cDNA Synthesis 

Total RNAs were extracted from the radicles of each replicate with the Spectrum™ Plant Total RNA Kit (Merck), scaling the reagent volumes recommended by the manufacturer to the low amount of root material per sample (12 mg on average), as follows: 300 µL of the Lysis Solution/2-ME Mixture, 500 µL of the Binding Solution, 300 µL for every washing step, and 2 subsequent elutions with 35 µL of the Elution Solution. Extracted RNA’s quantity was measured by Nanodrop 3.0 (Thermo Scientific), quality was assessed by 1% electrophoresis agarose gel, and integrity was measured by on-chip capillary electrophoresis using Agilent RNA 6000 Nano kit and Bio-Analyzer 2100 (Agilent Technologies, Boeblingen, Germany). Then, 1 µg of each RNA sample was purified from residual genomic DNA and reverse-transcribed to cDNA with Qiagen QuantiTect Reverse Transcription Kit, following the manufacturer’s instructions. RNA’s and cDNA’s yield and quality were estimated by Nanodrop, while the absence of residual genome traces was checked through qualitative PCR by amplifying the reference gene and using the following primer pair (GCCAGTGGACGCACAACAGGTA and CGCTCGGCAGTGGTGGTGAA for gene *Actin7*) designed on Primer3 to be on two contiguous exons. 

#### 4.2.4. Real Time qPCR

Real-time qPCR analysis was performed using the SsoFast EvaGreen Supermix (Bio-Rad, Hercules, CA, USA) and the CFX96 Real-Time PCR system (Bio-Rad Laboratories, Hercules, CA, USA). Each PCR reaction contained 10 μL of SsoFast EvaGreen Supermix, 10 µM of each primer, and 2 μL of cDNA (25 ng/μL) from each sample (final volume was 20 μL per reaction with sterile water). For each qPCR reaction, three technical replicas were produced. Real-time qPCR conditions were used as follows: 95 °C for 30 s; 35 cycles of 95 °C for 5 s; 58 °C for 5 s; the melting curve was assessed from 65 °C to 95 °C in increments of 0.5 °C. Standard curves for each primer pair and for each species were generated by plotting the quantification cycle (Cq) values from qPCRs executed with a pool of all cDNA samples as templates, as well as the log10 concentration of the cDNA template (5, 25, 50 and 100 ng/µL). The amplification efficiency (E) of each primer pair in each species was calculated from the slope of the corresponding standard curve as:$$E=10^{-1/\mathrm{slope}}$$
$$\%E=\left(E-1\right)\times 100$$
and ranged from 98 to 103% in *S. italica* and from 97 to 103% in *S. pumila*, with an average correlation value (R^2^) of 0.995.

Expression levels of the 7 target genes for the 24 cDNAs samples (12 for each species) were calculated as fold change:$$\mathrm{Fold}\;\mathrm{change}=2^{-\left(\Delta \Delta \mathrm{Cq}\right)}$$
where ΔΔCq value represents the difference between the average 1-h-self-DNA-treatment or the average 3-h-self-DNA-treatment ΔCq and the average control ΔCq. The average ΔCq values were calculated over the three biological replica ΔCq values, except for the control treatment. In this case, each gene average ΔCq was calculated over six biological replicates, given that the three replicates per exposure time were put together and assuming non-variation of gene expression under controlled conditions. 

#### 4.2.5. Statistical Analysis

To evaluate the effect of the DNA solution treatments on the target species root elongation, we fitted a factorial ANOVA model, including main and second order interactions of target species (S, two levels, *S. pumila* and *S. italica*): DNA source (D, four levels, *S. pumila*, *S. italica*, *B. napus*, and *S. salar*) and DNA concentration (C, three levels, 2, 10, and 50 ng/µL). The interaction terms were included in the model, considering that previous evidence [1] showed that self-DNA effects are species-specific, with magnitude depending on the target species’ sensitivity, DNA source, and concentration. Then, it was expected to observe significant S × D (due to species-specificity), S × C (due to species sensitivity), and D × C (due to different effects of different DNA sources at different concentration levels) terms. Root elongation data were further investigated with Tuckey’s test to assess the significance of pair-wise differences in the average root elongation percentage among all treatment groups (α = 0.05). We purposely decided to express response data for both target species as percentages of the respective controls in order to allow a comparison of the treatments’ effects between the two target species, while controlling for the different species-specific root elongations. Then, to assess the occurrence of significant differences in the comparisons between treatment groups and the respective controls, the average root elongation percentage of each treatment group was tested against the value 100 (i.e., the control mean) by one-sample *t* tests with the application of Bonferroni’s correction for multiple comparisons (α = 0.05/24). Borderline statistical significance was considered for tests producing marginal *p*-values (0.05 < *p* < α).

To evaluate the significance of the differences in average ΔCq values between 1 h and 3 h treatment and between 1 h treatment and control and 3 h treatment and control within each target gene, we carried out two independent-sample *t* tests with the application of Bonferroni’s correction for multiple comparisons (α = 0.05/21) for both species. 

Statistical analyses and graphs were performed using Excel 2013 (Microsoft Inc., Redmond, WA, USA), STATISTICA v. 10 (Statsoft Inc., Tulsa, OK, USA), and R software version 3.6.2 [64] using the following packages: base version 3.6.2, stats version 3.6.2, and ggplot2 version 3.2.1.

## 5. Conclusions

Our root inhibition provided confirmatory evidence on the concentration dependency and species-specificity of self-DNA inhibition. More importantly, the hypothesis that the self-DNA inhibitory effect still holds at infra-generic level was also confirmed for congeneric species with different ecological traits, such as the weedy invasive *S. pumila* and the cultivated *S. italica*. However, our research also highlighted some critical issues deserving verification by appropriate upscaled field tests, such as the extent of possible inhibition of crop species treated with DNA targeting closely related weeds. At the molecular level, among the 7 tested abiotic stress responsive genes, we found 4 and 5 genes differentially expressed in *S. italica* and *S. pumila*, respectively, after 1 and/or 3 h of exposure, supporting a previous indication of abiotic stress pathways’ involvement in the early response to self-DNA. In this respect, the main outcome of our qPCR experiment is the clear indication of the functional link between self-DNA exposure and ROS production at the early stage, as related to the enhancement of genes related to anti-oxidant activity. Finally, our exploratory molecular experiment should be followed by further tests addressing more specific cell processes with fully representative gene sets.

## Figures and Tables

**Figure 1 plants-12-01288-f001:**
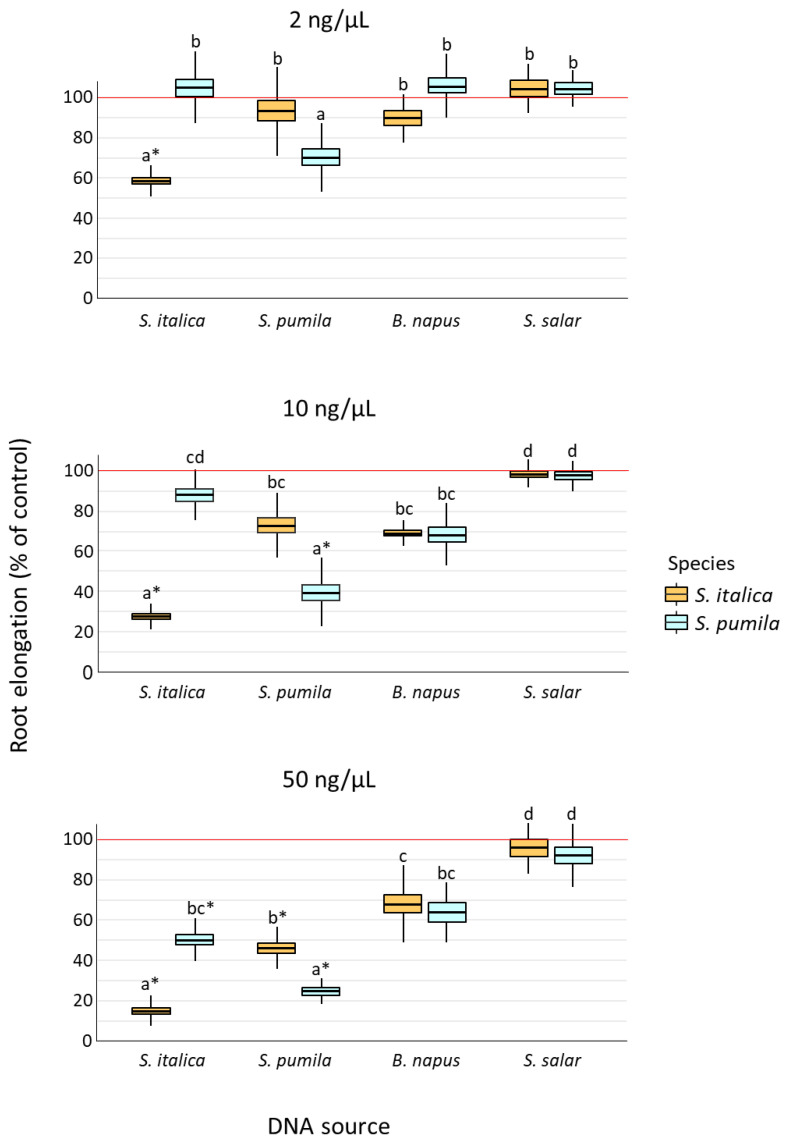
Effects of the treatment solutions containing DNA from different sources (*S. italica*, *S. pumila*, *B. napus*, *S. salar*) at three concentrations (2, 10, and 50 ng/µL) on the root elongation (% of control = 100, horizontal red lines) of *S. italica* and *S. pumila* seedlings after 4-day exposure in controlled conditions. Data refer to mean ± 1 standard error (box) and 95% confidence limits (whiskers) of 3 replicates for each treatment combination. Different letters above bars indicate significantly different means within each panel (Tuckey’s test, *p* < 0.05. Detailed results in Appendix A). Asterisks indicate root elongation inhibition as compared to the control (one-sample *t* test with Bonferroni’s correction for multiple comparisons).

**Figure 2 plants-12-01288-f002:**
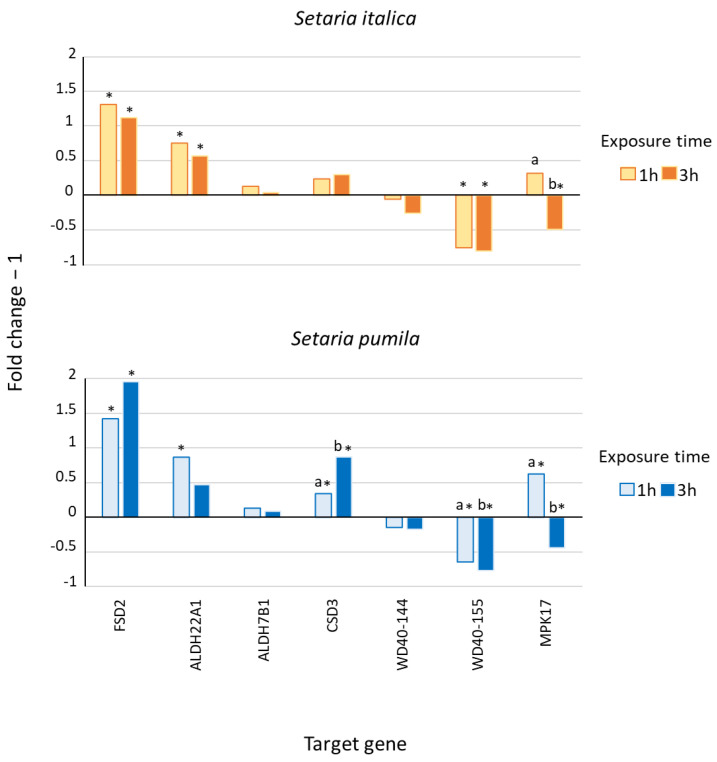
Target gene expression profiles in the two *Setaria* species after 1 and 3 h exposure to self-DNA. Data refer to fold change −1 for each target gene after exposure to ultra-purified self-DNA solutions, for 1 and 3 h, at the concentration of 90 ng/µL. Different letters above bars indicate statistically significant differences in ΔCq means between exposure times within each gene (*t* test for independent samples with Bonferroni’s correction for multiple comparisons). Asterisks indicate ΔCq means that are significantly different from the controls (*t* test for independent samples with Bonferroni’s correction for multiple comparisons. Detailed results in Appendix A).

**Figure 3 plants-12-01288-f003:**
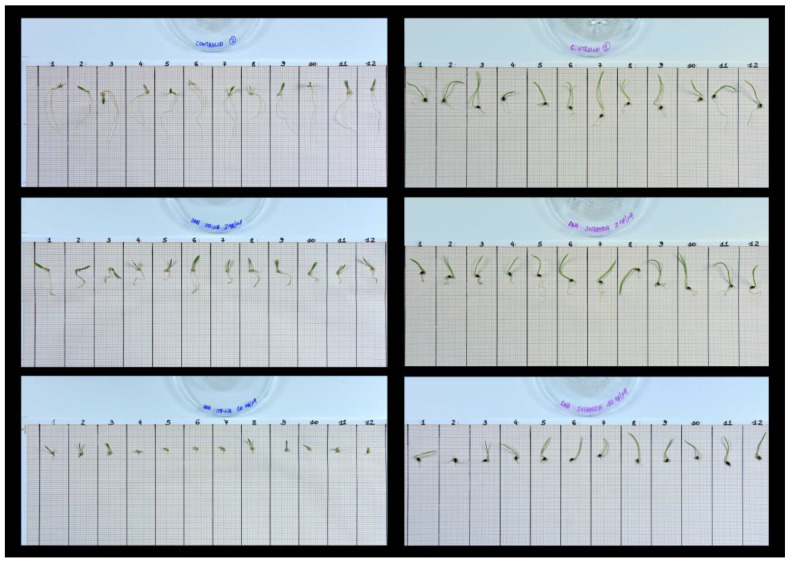
Examples of photographs of the seedlings from selected Petri dishes moved onto graph paper. Images refer to seedlings of *S. italica* (**left**) and *S. pumila* (**right**), unexposed (**top**) or exposed to self-DNA solutions, at lower (**center**) and higher (**bottom**) concentration.

**Table 1 plants-12-01288-t001:** Results of the ANOVA carried out on root elongation data from the cross-factorial experiment. Tested effects include main and second order effects of target species (S, two levels, *S. italica* and *S. pumila*), DNA source (D, four levels: *S. italica*, *S. pumila*, *B. napus*, *S. salar*), and concentration (C, three levels, 2, 10, and 50 ng/μL). Df = Degrees of freedom; SS = Sum of Squares; MS = Mean Sum of Squares; F = F statistic ratio; *p* = *p* value.

Effect	Df	SS	MS	F	*p*
Target species (S)	1	178.8	178.8	4.66	0.0353
DNA source (D)	3	24,862.5	8287.5	216.13	<0.0001
Concentration (C)	2	15,268.6	7634.3	199.09	<0.0001
S × D	3	13,069.7	4356.6	113.61	<0.0001
S × C	2	279.5	139.7	3.64	0.0328
D × C	6	3641.8	607.0	15.83	<0.0001
Error	54	2070.7	38.3		

## Data Availability

Not applicable.

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
