# Peer review of "Self-DNA Early Exposure in Cultivated and Weedy Setaria Triggers ROS Degradation Signaling Pathways and Root Growth Inhibition"

_plants, 2023, doi:10.3390/plants12061288_

Round 1

Reviewer 1 Report

This is an interesting study, worth publishing and generally well-written. However, there are a few aspects that need improvement or correction, mostly related to the statistical analyses:

Lines 480-511: the authors describe how they designed the primers to be used in PCR, but they do not specificy the source of the primers used (after design, how where the custom primers sourced?)

Lines 569-570: a three-way ANOVA model  with interactions was performed”. Please clarify whether all interactions were included in the model or only partial interactions (if the latter which ones?).

Lines 575-577: „The average root elongation percentage of each treatment group  was also compared to the control growth through one-sample t tests with the application  of Bonferroni’s correction for multiple comparisons (α = 0.05/24)”. Bonferroni correction is indeed, conservative, but it is not clear to us why a similar ANOVA plus Tukey (or plus Bonferroni for that matter) was not used. The same questions also hold for the next paragraph.

Lines 590-592: please clarify whether any particular packages were used for the analyses.

In our view, Fig. 1 could rather use boxplots or violinplots, which would increase the visual quality of the representation.

Lines 98-100: “Target species (S) was not a significant factor in the ANOVA model, meaning  that the two Setaria species responded similarly to the treatments, contrary to what expected”. Actually, this is statement is somewhat dubious, considering that the authors report significant interactions S x D x C. It is not clear to me how large is this interaction effect, and whether a simpler model excluding this interaction would not be more helpful. But if the authors prefer to use this model (I do not know on what basis they decided to add all the interaction terms), they should at least use a plot to allow the reader to grasp the sense of the interaction (although I must acknowledge that I would rather be tempted to use a simpler model with only binary interactions, i.e. excluding this S X D X C interaction). Unless they have field knowledge that S*C and S*D interactions are important, I would also rather recommend them to exclude these two interactions (because they are anyway not significant). They would result in simpler models and easier to interpret. Finally, for the D x C interaction it would be important for the authors to show an interaction plot, so as the reader is able to easily understand the sense of the interaction.

Lines 138-140: “Such pattern was consistent with the statistically significant interactive effect of target species and  DNA specificity (S × D, P < 0.0001) “. As a matter of fact, table 2 has a contents identical with table 1 and needs to be replaced (as the current value from table 2 states p = 0.2868).

Reviewer 2 Report

The manuscript can be of interest to wide readers of Journals and contributes to existing knowledge on the subject matter. However, I have pointed out few pertinent points for improving the clarity of the content and boosting the scientific soundness of the manuscript.

Introduction

More information may be added on pertinence of Setaria.

Information is insufficient on peer-findings for highlighting the research and knowledge gaps

Remove e.g.; from the line …[e.g.,10], and add more citations, as you mentioned “some previous papers”

Line 124, 126, 151, 163, : Write in italic form

(S. italica, S. pumila, B. napus, S. salar)

Results and discussions

This is well written, however Authors need to strengthen the discussion section by adding more interpretations of recorded findings supported by peer-findings.

Line 332: Check the spelling of “mais (Zea mays L.)”

Line 442: Write the proper address of equipment you used in the experiments (model, make, Adress) ” Nanodrop spectrophotometer.”

Line 472: Check the spelling …………….”ImageJ”

Line 471-472: where are the images?  “The images obtained at the initial and final observation stages were analysed with the ImageJ software

Reviewer 3 Report

I read the research carefully and found that the experiments are well organized and implemented and the topic is very interesting and has novelty, but minor improvements are needed. Overall, the manuscript meets publication criteria for the journal. However, in the Materials and Methods section, line 439: There are two citations that are not in the citation style of the journal. The conclusion section should be shortened to one paragraph of 3 or 4 short sentences.

Reviewer 4 Report

The manuscript by Ronchi et al. focuses on a novel attention-grabbing topic with emerging applications. Overall the study seems not properly described, with the methodologies substantially lacking in many points. Although it is a novel field of research, the authors should improve the robustness of the experiments before the manuscript can be considered for publication.

Major flaws:

Looking at the results in figure 1 it is very obvious that it is not possible to exclude that the inhibitory effect is related to the DNA extraction procedure and the carryover effect of possible inhibiting compounds for DNA from S. italica and S. pumila. This holds either for plant compounds extracted with plant DNA or for extraction reagents. Diluting the used DNA solutions (2ng/ul) the authors obtained a better root elongation which can be an effect of diluting inhibiting compounds.

The manual extraction protocol described is widely known to retain some inhibiting compounds (polyphenols) to the final steps. This risk is also confirmed by the very low inhibiting effect, instead shown using commercial Salmon salar DNA solution. 

From chapter 4.11.3 is not clear how the B. napus DNA was obtained. Was it sonicated and purified as well?

To improve the robustness of the experiment, I would suggest using only purified commercial DNA when available. In other cases, I would suggest comparing the effect of DNA extracted with different methodologies. As an example, I would add to the CTAB method an extraction with silica column kit(s) to exclude the possible carry-over effect.

What are "pre-germinated seeds"? Are they seedlings? Seeds are imbibed in water (or other solutions) and they complete the germination process when the radicle protrudes.

Lane 457-458, please explain what the “pre-germination phase” is. Do you mean "seed imbibition"?

Besides the RNA yields obtained the authors should show the RNA integrity values (RIN or RIS) which are extremely important to carry out a gene expression analysis using representative samples.

The M&M description is verbose. The same passages are reported in different paragraphs.

The authors should also present the different effects by showing photographs of the root elongation they took.

Round 2

Reviewer 2 Report

Author has revised the manuscript and responded to all the comments.

Now, the manuscript can be accepted in its present form.

Author Response

Author has revised the manuscript and responded to all the comments. Now, the manuscript can be accepted in its present form.

We thank the Reviewer for his/her appreciation of our study as well as for his/her suggestion of acceptance for publication in Plants. 

Reviewer 4 Report

The new version of the manuscript clearly presents significant improvements with respect to the original version. However, there are still fundamental flaws that make the study still not enough scientifically solid. The absorbance ratios 260/280 of 0.97 and 0.78 obtained (lane 509) for DNA extracted from S. italica and S. pumila, respectively, is just an unpublishable result in the “Plants” journal with a high IF and relevant position in the Plant Science category. Abnormal low 260/280 ratios usually indicate a high concentration of phenols in the sample which can derive from residual phenol (during the separation phase in the extraction protocol) or polyphenolic substances naturally present in the sample, which should be removed (using PVP for instance..). I agree with the authors (lane 507) that stated that these are poor-quality spectrophotometric values and I think that these DNA extractions should be not used as a basis for downstream experiments. The authors should start with more reliable DNA extractions (there are hundreds of DNA extraction protocols in the literature). Using such low-quality DNA the conclusion that “there is a self-DNA inhibitory effect” does not hold.

Round 3

Reviewer 4 Report

The authors present an improved version of the manuscript. The study appears now more solid. I suggest the authors to move the figure S3 into the main text improving the quality of the image.

Author Response

We thank the referee for acknowledging our effort to improve the manuscript. Following his/her suggestion, we have now moved the Figure S3 into the main text (new Figure 3) producing a higher-quality version.